# Dolores: Deep Contextualized Knowledge Graph Embeddings

**Haoyu Wang**                                          WHY16GZL@SEAS.UPENN.EDU
*University of Pennsylvania*

**Vivek Kulkarni**                                          VIVEKSCK@STANFORD.EDU
*Stanford University*

**William Yang Wang**                                          WILLIAM@CS.UCSB.EDU
*University of California, Santa Barbara*

## Abstract

We introduce DOLORES, a new knowledge graph embeddings, that effectively captures contextual cues and dependencies among entities and relations. First, we note that short paths on knowledge graphs comprising of chains[1] of entities and relations can encode valuable information regarding their contextual usage. We operationalize this notion by representing knowledge graphs not as a collection of triples but as a collection of entity-relation chains, and learn embeddings using deep neural models that capture such contextual usage. Based on BiLSTMs, our model learns deep representations from constructed entity-relation chains. We show that these representations can be easily incorporated into existing models to significantly advance the performance on several knowledge graph tasks like link prediction, triple classification, and multi-hop knowledge base completion.

## 1. Introduction

Knowledge graphs [Dong et al., 2014] enable structured access to world knowledge and form a key component of several applications like search engines, question answering systems and conversational assistants. Knowledge graphs are typically interpreted as triples of the form (`entityA`, `relationX`, `entityB`), representing a relation (`relationX`) between `entityA` and `entityB`. However, a limitation of using discrete representations of triples is that it does not easily enable one to infer similarities and potential relations among entities which may be missing in the knowledge graph. Consequently, one popular alternative is to learn dense continuous representations of entities and relations by embedding them in latent continuous vector spaces, while seeking to model the inherent structure of the knowledge graph. Most knowledge graph embedding methods can be classified into two major classes [Wang et al., 2017]: one class which operates purely on triples like RESCAL [Nickel et al., 2011], TRANSE [Bordes et al., 2013], DISTMULT [Yang et al., 2015], TRANSD [Ji et al., 2015], COMPLEX [Trouillon et al., 2016], CONVE [Dettmers et al., 2018] and the second class which seeks to incorporate additional information (like multi-hops). Learning high-quality knowledge graph embeddings can be quite challenging given that (a) they need to effectively model the contextual usages of entities and relations and (b) they would need to be useful for a variety of predictive tasks on knowledge graphs.

---

[1]We use the word "path" when we talk about a graph and use "chain" when comparing with an entity-relation pair or triple. Essentially, the two terms share the same meaning.

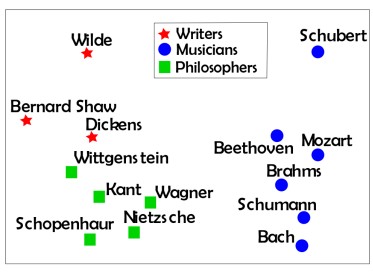 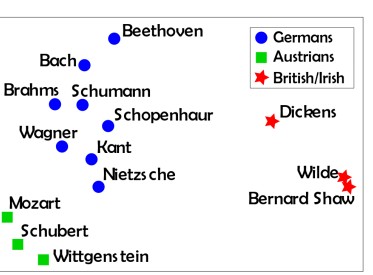 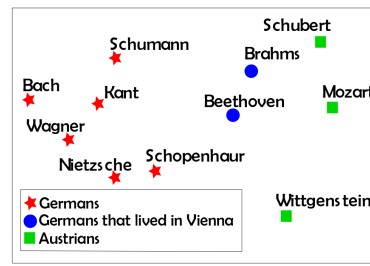

(a) Context-independent     (b) Nationality dependent     (c) Place-lived dependent

Figure 1: Context independent and dependent embeddings learned by DOLORES. Figure (a) shows context-independent representations of writers (red), philosophers (green), and musicians (blue); Figure (b) shows contextual embeddings with relation "/people/nationality": Austrians (green), Germans (blue), British/Irish (red). Figure (c) shows contextual embeddings with relation "/people/place_lived/location": we can see that Beethoven and Brahms (blue), though Germans (other Germans are in red), once lived in Vienna, Austria (Austrians are in green), and their embeddings lie in between those of Germans and Austrians.

In this paper, we present DOLORES, a new type of knowledge graph embeddings that are *deep* and *contextualized*. Both context-independent and context-dependent embeddings of entities and relations are learned through a deep neural sequential model and then used to compute DOLORES. Figure 1 illustrates the learned representations: note that the contextual independent entity embeddings (see Figure 1a) reveal three clusters of entities: writers, philosophers, and musicians. While the contextual dependent embeddings in turn effectively take on different clusterings and account for specific relations (nationality and place-lived).

Inspired by recent advances in learning word representations from deep neural language models [Peters et al., 2018] using Bi-Directional Long Short-Term Memory (BiLSTM), we derive connections between language models and *random walks on knowledge graphs*. These connections enable us to propose new "deep contextualized" knowledge graph embeddings. Unlike most knowledge graph embeddings such as TransD, TransE [Bordes et al., 2013, Ji et al., 2015] etc. which are learned using relatively shallow models, the representations from DOLORES are deep. They are functions of internal states of a BiLSTM [Hochreiter and Schmidhuber, 1997] accepting an entire path as input. Furthermore, they are composed of representations learned at various layers, potentially capturing different degrees of abstraction.

DOLORES can be used as input representations for predictive models on knowledge graphs. As long as a predictive model uses input representations for entities and relations, we can easily replace those representations with DOLORES and significantly improve its performance. Specifically, we show that DOLORES advances neural models on various tasks like link prediction, triple classification and multi-hop knowledge base completion.

To summarize, our contributions are as follows:

1. We present a new method of learning deep contextualized knowledge graph embeddings using a deep neural sequential model.

2. These embeddings are functions of hidden states of the deep neural model and can capture both *context-independent* and *context-dependent* cues.

3. We show empirically that Dolores can easily be incorporated into existing predictive models on knowledge graphs to advance performances on several tasks like link prediction, triple classification, and multi-hop knowledge base completion.

## 2. Related Work

Extensive work exists on knowledge graph embeddings dating back to Rescal [Nickel et al., 2011] which is based on a matrix factorization approach. Bordes et al. [2013] advanced this line of work by proposing the first translational model, TransE, which seeks to relate the head and tail entity embeddings by modeling relations as translational vectors. This culminated in a long series of new knowledge graph embeddings, all based on the translational principle with various refinements [Wang et al., 2014, Lin et al., 2015b, Ji et al., 2015, Yang et al., 2015, Trouillon et al., 2016, Nickel and Kiela, 2017, Minervini et al., 2017, Xiao et al., 2017, Ma et al., 2017, Chen and Zaniolo, 2017, Chen et al., 2018]. Some recently proposed models like ManiFoldE [Xiao et al., 2016] attempt to learn knowledge graph embeddings as a manifold while embeddings like HolE [Nickel and Kiela, 2017] derive inspiration from associative memories.

With the success of neural models, a few approaches based on convolutional neural networks have been proposed to learn knowledge graph embeddings [Dettmers et al., 2018, Shi and Weninger, 2017, Nguyen et al., 2018a]. There has been some work on incorporating additional information like entity types, relation paths etc. to learn knowledge graph representations. Palumbo et al. [2018] used node2vec to learn embeddings of entities in a knowledge graph. A notable class of methods called "path-ranking" based models directly consider paths between entities as features. Examples include Path Ranking Algorithm (PRA) [Lao et al., 2012], PTransE [Lin et al., 2015a] and models based on recurrent neural networks [Neelakantan et al., 2015, Das et al., 2017, Yin et al., 2018]. Besides, generative adversarial networks was employed to address the problem [Cai and Wang, 2018] and Das et al. [2018] proposed a reinforcement learning method that deals with practical tasks of answering questions where the relation is known, but only one entity. Hartford et al. [2018] modeled interactions across two or more sets of objects using a parameter-sharing scheme.

While most of the aforementioned models are shallow (except for part of the recurrent neural network models), our model Dolores differs from all of these works and especially that of Palumbo et al. [2018], in that we learn deep contextualized knowledge graph representations of entities and relations using a deep neural sequential model. The work that is relatively close to ours is that of Das et al. [2017] and Yin et al. [2018] who used RNN-based architectures to model paths. However, our focus is not on path reasoning but on learning rich knowledge graph embeddings useful for a variety of predictive tasks. Moreover while Das et al. [2017] needed to use paths generated from PRA that typically correlate with relations, our method has no such restriction and only uses paths obtained by generic random walks, which greatly enhancing the scalability of our method.

## 3. Method and Models

### 3.1 Problem Formulation

Given a knowledge graph $G = (E, R)$ where $E$ denotes the set of entities and $R$ denotes the set of relations among those entities, we seek to learn $d$-dimensional embeddings of the entities and relations. In contrast to previous knowledge graph embedding methods like [Bordes et al., 2013, Wang et al., 2014, Ji et al., 2015, Lin et al., 2015b, Trouillon et al., 2016] which are based on shallow models and operates primarily on triples, we use a deep neural model to learn *"deep"* and *"contextualized"* knowledge graph embeddings. Our method consists of two main components:

1. **Path Generator**

   This component is responsible for generating a large set of entity-relation chains that reflect the varying contextual usages of entities and relations in the knowledge graph.

2. **Learner**

   This component is a deep neural model that takes entity-relation chains as input and learns entity and relation embeddings which are weighted linear combination of internal states of the model thus capturing context dependency.

We highlight the key connections that enable us to build on recent advances to learn a new knowledge graph embeddings in the following section.

### 3.2 Preliminaries

**Language Modeling**    Recall that the goal of a language model is to estimate the likelihood of a sequence of words: $w_1, w_2, \cdots, w_n$ where each word $w_i$ is from a finite vocabulary $\mathcal{V}$. Specifically, the goal of a *forward language model* is to model $\Pr(w_i|w_1, w_2, \cdots, w_{i-1})$. While traditionally this has been modeled using "n-gram" based models [Manning and Schutze, 1999, Jurafsky, 2000], deep neural networks like LSTMs and RNNs have been used recently. As noted by Peters et al. [2018], a forward language model using an LSTM of $L$ layers works as follows: at position $k$, each LSTM layer outputs a *context-dependent* representation denoted by $\overrightarrow{h_{k,j}}$ corresponding to the $j^{th}$ layer of the LSTM. The top-most layer of the LSTM is then fed as input to a softmax layer of size $|\mathcal{V}|$ to predict the next token. Similarly, a *backward language model* which models $\Pr(w_i|w_{i+1}, w_{i+2}, \cdots, w_n)$ can be implemented using a *"backward"* LSTM producing similar representations, denoted $\overleftarrow{h_{k,j}}$ this time. A BiLSTM just combines both directions and seeks to jointly maximize the log-likelihood of the forward and backward directional language model objectives.

These context-dependent representations learned by the LSTM at each layer have shown to be useful as "deep contextual" word representations in various predictive tasks in natural language processing [Peters et al., 2018]. In line with this trend, we will also use deep neural sequential models to learn DOLORES.

**Connection between Random Walks on Graphs and Sentences in Language** Since the input to a language model is a large corpus or set of sentences, one can generalize language modeling approaches to graphs by noting that the analog of a sentence in graphs is a

"random walk". More specifically, a truncated random walk of length $T$ containing a node $v$ is analogous to a sentence and effectively captures the "context" of $v$ in the graph. Since the power-law distribution of vertices appearing in random walks follows the same distribution of words in natural language [Perozzi et al., 2014], we consider an $(entity, relation)$ pair, rather than an entity or a relation, as a "word" in language models.

In particular, a random walk on a knowledge graph starting at entity $e_1$ and ending at entity $e_k$ is a sequence of the form $e_1, r_1, e_2, r_2, \cdots, e_k$ representing the entities and the corresponding relations between $e_1$ and $e_k$ in the knowledge graph. Moving forward we denote such a path comprising of entities and relations by $\mathbf{q} = (e_1, r_1, e_2, r_2, \cdots, e_k)$. We generate a large set of such paths from the knowledge graph $G$ by performing several random walks on it which in turn yields a corpus of "sentences" $S$ needed for "language modeling".

### 3.3 Dolores: Path Generator

As discussed in preliminaries, we denote $S$ as the set of paths obtained by doing random walks in a knowledge graph, adopting one component of NODE2VEC [Grover and Leskovec, 2016] to construct $S$. In particular, we perform a $2^{nd}$ order random walk with two parameters $p$ and $q$ that determine the degree of breadth-first sampling and depth-first sampling. Specifically, $p$ controls the likelihood of immediately revisiting a node in the walk whereas $q$ controls whether the walk is biased towards nodes close to or away from the starting node. We emphasize that while NODE2VEC has additional steps to learn dense continuous representations of nodes, we adopt only its first part to generate a corpus of random walks representing paths in knowledge graphs.

### 3.4 Dolores: Learner

While traditional language models treat a sentence as a sequence of words, we adopt the same machinery to model a path consisting of entities and relations to obtain knowledge graph embeddings. Given paths generated from the path generator component, we would like to model the probability of an entity-relation pair given the history and future context via a BiLSTM. In particular, the forward direction LSTM models:

$$\Pr([e_1, r_1], [e_2, r_2], \cdots, [e_N, r_N]) =$$

$$\prod_{t=1}^{N} \Pr([e_t, r_t] \mid [e_1, r_1], [e_2, r_2], \cdots, [e_{t-1}, r_{t-1}]). \tag{1}$$

Similarly, the backward direction LSTM models:

$$\Pr([e_1, r_1], [e_2, r_2], \cdots, [e_N, r_N]) =$$

$$\prod_{t=1}^{N} \Pr([e_t, r_t] \mid [e_{t+1}, r_{t+1}], \cdots, [e_N, r_N]). \tag{2}$$

Figure 2 illustrates this succinctly. At each time-step $t$, we deal with an entity-relation pair $[e_t, r_t]$. We first map one-hot vectors of $e_t$ and $r_t$ using an embedding layer, and then concatenate the two mapped vectors to obtain context-independent representations which

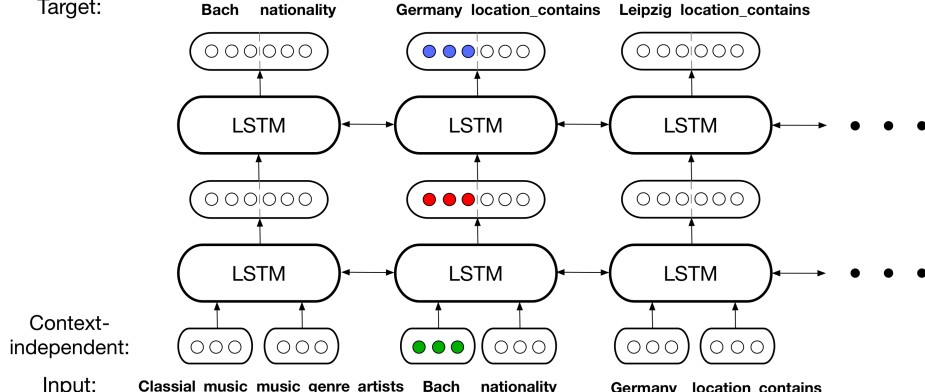

Figure 2: Unrolled RNN architecture. The input to the deep BiLSTM is entity-relation chains generated from random walks in KG. At each time step, the LSTM consumes the concatenation of entity and relation vectors. The "target" shown in the figure is the training target for the forward direction. Learned contextual representation of entity "**Bach**" is the concatenation of the embedding layer (green) and a linear combination of internal states of deeper layers (red and blue).

are then passed through $L$ layers of a BiLSTM. Each layer of LSTM outputs the pair's 2-directional context-dependent representation $\overrightarrow{h_{t,i}}$ and $\overleftarrow{h_{t,i}}$, where $i=1, 2, \cdots, L$. Finally, the concatenation of output of the top layer, $[\overrightarrow{h_{t,L}}, \overleftarrow{h_{t,L}}]$, is used to predict the next pair $[e_{t+1}, r_{t+1}]$ and $[e_{t-1}, r_{t-1}]$ respectively using a softmax layer.

$$\sum_{t=1}^{N} \log \Pr([e_t, r_t] \mid [e_1, r_1], \cdots, [e_{t-1}, r_{t-1}]; \boldsymbol{\Theta_F})+$$
$$\sum_{t=1}^{N} \log \Pr([e_t, r_t] \mid [e_{t+1}, r_{t+1}], \cdots, [e_N, r_N]; \boldsymbol{\Theta_B}), \tag{3}$$

We jointly maximize the log likelihood of the forward and backward directions as shown in Equation 3 where $\boldsymbol{\Theta_F}=(\theta_x, \overrightarrow{\theta_{LSTM}}, \theta_s)$ respectively corresponds to the parameters of the embedding layer, forward-direction LSTM and the softmax layer; $\boldsymbol{\Theta_B}=(\theta_x, \overleftarrow{\theta_{LSTM}}, \theta_s)$ corresponds to the similar set of parameters for the backward direction. Specifically, we share the parameters for the embedding and softmax layer across both directions. We maximize Equation 3 by training the BiLSTMs using back-propagation.

**Extracting Dolores from the learner**  Having estimated the parameters of the DOLORES learner, we now extract the context-independent and context-dependent representations for each entity and relation, and combine them to obtain DOLORES. More specifically, DOLORES are *task-specific combinations* of the context-dependent and context-independent representations. By accepting triples or paths from certain tasks as input (not the paths generated by the path generator), our learner (which is an $L$-layer BiLSTM) computes a set of $2L+1$ representations for each entity-relation pair which we denote by:

$$R_t = [x_t, \overrightarrow{h_{t,i}}, \overleftarrow{h_{t,i}} \mid i = 1, 2, \cdots, L],$$

| Task | Baseline | | Dolores+ Baseline | Increase (Absolute/ Relative) |
|---|---|---|---|---|
| Link Prediction (head) | [Nguyen et al., 2018b] | 15.7 | 18.7 | 3.0 / 3.56% |
| Link Prediction (tail) | [Nguyen et al., 2018b] | 32.8 | 37.2 | 4.4 / 6.55% |
| Triple Classification | [Nguyen et al., 2018b] | 87.00 | 87.55 | 0.55 / 4.23% |
| Multi-hop KB Completion | [Yin et al., 2018] | 76.16 | 78.28 | 2.12 / 8.9% |

Table 1: Summary of results of incorporating Dolores into existing models for various tasks. Note that in each case, simply incorporating Dolores results in an improvement of performance (relative error reduction over the baseline) in various tasks like link prediction, triple classification and multi-hop knowledge base completion.

| | FB15K237 | | | | | | | | |
|---|---|---|---|---|---|---|---|---|---|
| | HEAD | | | TAIL | | | Avg. | | |
| Method | MRR | MR | HITS@10 | MRR | MR | HITS@10 | MRR | MR | HITS@10 |
| TransE | 0.154 | 651 | 0.294 | 0.332 | 391 | 0.524 | 0.243 | 521 | 0.409 |
| PTransE (RNN, 2-step) | 0.172 | 454 | 0.335 | 0.354 | 298 | 0.544 | 0.263 | 376 | 0.440 |
| RNN-Path-entity | 0.198 | 392 | 0.363 | 0.375 | 204 | 0.579 | 0.287 | 298 | 0.471 |
| ConvKB | 0.157 | 426 | 0.300 | 0.328 | 193 | 0.543 | 0.243 | 309 | 0.515 |
| ConvKB (+ Dolores) | 0.187 | 402 | 0.352 | 0.372 | 180 | 0.577 | 0.280 | 291 | 0.465 |
| Improvement (relative %) | 19.11% | 5.63% | 17.33% | 13.41% | 6.74% | 6.26% | 15.23% | 5.83% | 10.19% |

Table 2: Performance of incorporating Dolores into an existing model, ConvKB, for link prediction. Note that DOLORES significantly improves the performance of the baseline model on various sub-tasks: head entity and tail entity prediction. *For all metrics except MR (mean rank) higher is better.*

where $x_t$ is the context-independent embedding and $\overrightarrow{h_{t,i}}, \overleftarrow{h_{t,i}}$ correspond to the context-dependent embeddings from layer $i$.

Given a downstream model, which is not necessarily the pre-trained LSTM, Dolores learns a weighted linear combination of the components of $R_t$ to yield a single vector for usage in the embedding layer of the downstream model. In particular

$$\text{Dolores}_t = [x_t, \sum_{i=1}^{L} \lambda_i \cdot h_{t,i}], \tag{4}$$

where we denote $h_{t,i} = [\ \overrightarrow{h_{t,i}},\ \overleftarrow{h_{t,i}}\ ]$ and $\lambda_i$'s denote *task-specific* learnable weights of the linear combination.

**Incorporating Dolores into existing predictive models on Knowledge Graphs** While it is obvious that our embeddings can be used as features for new predictive models, it is quite easy to incorporate our learned Dolores into existing predictive models on knowledge graphs. The only requirement is that the model accepts as input, an embedding layer (for entities and relations). If a model fulfills this requirement (which a large number of neural models on knowledge graphs do), we can just use Dolores as a drop-in replacement by initializing the corresponding embedding layer with Dolores. In our evaluation below, we show how to improve several neural models on various tasks by incorporating Dolores as a drop-in replacement to the original embedding layer.

## 4. Experiments

We evaluate Dolores on 4 prediction tasks on knowledge graphs. In each case, adding Dolores to existing neural models improves performance significantly which we demonstrate in Table 1. While we primarily show that we can advance the model by incorporating Dolores as a "drop-in" replacement, for the sake of completeness, we also report numbers of other strong baseline methods to put the results into context. In the remainder of the section, we briefly provide high-level overviews of each task and summarize results for all tasks considered.

### 4.1 Experimental Settings for Dolores

Here we outline our model settings for learning Dolores. We generate 20 chains[2] for each node in the knowledge graph with the length of each chain being 21 (10 relations and 11 entities appear alternately). Our model uses $L = 4$ LSTM layers with 512 units and 32 dimension projections. We use residual connections between layers and the batch size is set to 1024 during the training process. We train Dolores for 200 epochs on corresponding datasets while variational dropout of 10% is added to LSTM hidden layers. Finally, we use Adam as the optimizer with appropriately chosen learning rates based on validation sets.

### 4.2 Evaluation Tasks

We consider three tasks, link prediction, triple classification, and multi-hop knowledge base completion [Das et al., 2017]. For the significance test, we used bootstrap resampling method.[3]

**Link Prediction** A common task in knowledge graph is link prediction, aiming to predict the missing entity when the other two parts of a triplet (*h, r, t*) are given. In-line with prior work [Dettmers et al., 2018], we carry out experiments on the FB15K-237 and WN18RR dataset in the filtered setting. Our results on FB15K-237 in terms of Mean Reciprocal Rank (MRR), Mean Rank (MR) and Hits@10 are shown in Table 2. Note that ConvKB[4] achieves an MRR of $(0.157, 0.328)$ on the head and tail link prediction tasks respectively. Observe that incorporating Dolores significantly improves the head and tail entity prediction performance by 19.11% and 13.41% respectively. Similar improvements are also observed on other metrics like Mean Rank (MR: lower is better) and Hits@10 (higher is better).

**Triple Classification** The task of triple classification consists in predicting whether a triple (*h, r, t*) is correct or not. Triple classification is a binary classification task widely explored by previous work [Bordes et al., 2013, Wang et al., 2014, Lin et al., 2015b]. We choose WN11 and FB13, two most frequently used benchmark datasets containing negative triples, and report the results of our evaluation in Table 3. When adding Dolores to ConvKB, we observe that the average performance is improved by 0.55 points (from 87.00 to 87.55).

---

[2]The total number of chains generated for the training, development and test set is approximately $300K$. Also, we observed no significant difference when using larger number of chains.

[3]All **bold** results in this paper are the best scores, denoting better results with the significance level $p$ <0.05 comparing with corresponding baselines.

[4]Upon the submission of this paper to AKBC, we did not notice the evaluation problem that arises in ConvKB's implementation. We have re-run our experiments and got numbers different from results in the previous version of this paper.

| Model | WN11 | FB13 | Avg. |
|---|---|---|---|
| NTN | 70.6 | 87.2 | 78.90 |
| TransE | 86.5 | 87.5 | 87.00 |
| TransD | 86.4 | 89.1 | 87.75 |
| PTransE (RNN, 2-step) | 86.9 | 88.1 | 87.50 |
| RNN-Path-entity | 86.5 | 87.9 | 87.20 |
| ConvKB | 86.4 | 87.5 | 87.00 |
| ConvKB (+ Dolores) | 86.9 | 88.2 | 87.55 |

Table 3: Accuracy results (in %) of triple classification on WN11 and FB13 test sets. The results of NTN [Socher et al., 2013], TransE [Bordes et al., 2013], TransD [Ji et al., 2015], ConvKB are from [Nguyen et al., 2018a]. Figures of PTransE [Lin et al., 2015a] and RNN-Path-entity [Das et al., 2017] are from our in-house implementation.

Though in terms of absolute value the improvement is small, when compared with the neural method NTN [Socher et al., 2013], or the path model RNN-Path-entity [Das et al., 2017], the improvement is evident.

**Multi-hop Knowledge Base Completion** The goal of the third task is to reason on the paths connecting an entity pair to predict missing relation types. We follow Das et al. [2017] and use the same dataset released by Neelakantan et al. [2015] which is a subset of FreeBase enriched with information from ClueWeb. The dataset consists of a set of triples $(e_1, r, e_2)$ and the set of paths connecting the entity pair $(e_1, e_2)$ in the knowledge graph. These triples are collected from ClueWeb by considering sentences that contain the entity pair in FreeBase. Neelakantan et al. [2015] inferred the relation types by examining phrases between pairs of entities. We use the same evaluation criterion as used by Das et al. [2017] and report our results in Table 4. Note that adding Dolores to the model from Yin et al. [2018] yields an 8.9% improvement (relative error reduction over 76.16).

| Model | MAP |
|---|---|
| PRA | 64.43 |
| PRA + Bigram | 64.93 |
| RNN-Path-entity | 71.74 |
| ROP_ARC1 | 74.23 |
| ROP_ARC2 | 74.46 |
| ROP_ARC3 | 76.16 |
| RNN-Path-entity (+Dolores) | 74.42 |
| ROP_ARC3 (+Dolores) | 78.28 |

Table 4: Results of multi-hop knowledge base completion. Numbers of PRA [Lao et al., 2011], PRA + Bigram [Neelakantan et al., 2015], and RNN-Path-entity [Das et al., 2017] are from the original paper. It yields an improvement of 8.9% (76.16 vs 78.28) on mean average precision (MAP) when ROP_ARC3 [Yin et al., 2018] is incorporated with Dolores.

Altogether, viewing the results on various tasks holistically, we conclude that incorporating Dolores into existing models improves their performance on each of these tasks, which

suggests that our embeddings can be effective in yielding performance gains on a variety of predictive tasks.

## 5. Analysis

| Model | Source | Nearest Neighbor(s) |
|---|---|---|
| TransE | Berlin | Munich, Stuttgart, Paris, Vienna, Free University of Berlin |
| RNN-Path-entity | Mendelssohn, place-lived, Berlin, capital-of, Germany | Claude Monet, place-born, Paris, place-lived, Ernest Hemingway |
| | Great Britain, participating-country, 1936-Olympics, event-location, Berlin | Boxing, Olympic-sport, 2004-Olympics, event-location, Athens |
| Dolores | Mendelssohn, place-lived, Berlin, capital-of, Germany | Bertolt Brecht, place-of-death, Berlin, adjoins, Brandenburg |
| | Great Britain, participating-country, 1936-Olympics, event-location, Berlin | Brazil, participating-country, 2012-Olympics, event-location, London |

Table 5: Nearest neighbors to "Berlin" using TransE, RNN-Path-entity and Dolores in the context of entity-relation chains ("Berlin" is weighted more in the similarity metric).

Given the improvement Dolores brings about in various tasks, the contextualized representations must capture information that cannot be seen by using common methodologies. As Table 5 shows, the nearest neighbors of "Berlin" using TransE [Bordes et al., 2013] method are related cities or universities like "Munich" and "Free University of Berlin"; the chains found by RNN-Path-entity [Das et al., 2017] notice only the relation ("*place-lived*" and "*event-location*") before the pivotal token; whereas Dolores is paying attention to a longer (or the whole) context, finding out counterparts for the token (e.g., "*Berlin*" related with both a person and a place in the first case; "*London*" as a host city of an Olympics with participating country in the second case). We believe it is the contextual dependencies captured by our embeddings that bring about improvement in tasks like multi-hop knowledge base completion.

**Error analysis of link prediction task** We analyze the predictions made when using Dolores on link prediction tasks by grouping test triples by the first component of their relation (the most abstract level) and computing the mean rank of the tail entity output over each group. We compare this metric against what TransE [Bordes et al., 2013] and ConvE [Dettmers et al., 2018] obtains. In particular, Figure 3a shows the categories where Dolores performs the best. To illustrate this, we show a few instances of such triples below. Note that when the head entity is a very specific entity like "*Louis Costello*", our method is very accurate at predicting the correct tail entity (in this case "*Comedian*").

- *(A serious man, film-release-region, Hong Kong)*

- *(Cabaret, film-genre, Film Adaptation)*

- *(Louis Costello, people-profession, Comedian)*

Our method is able to model such cases better because for specific entities, Dolores have captured the rich context associated with them from the entire paths. In contrast, Figure 3b shows the relation categories that Dolores performs relatively worse, which are basically broad relation categories, e.g., "base", "media-common". We list a few triples as follows:

- *(Psychology, students-majoring, Yanni)*

- *(Decca Records, music-record-label-artist, Jesseye Norman)*

- *(Priority Records, music-record-label-artist, Carole King)*

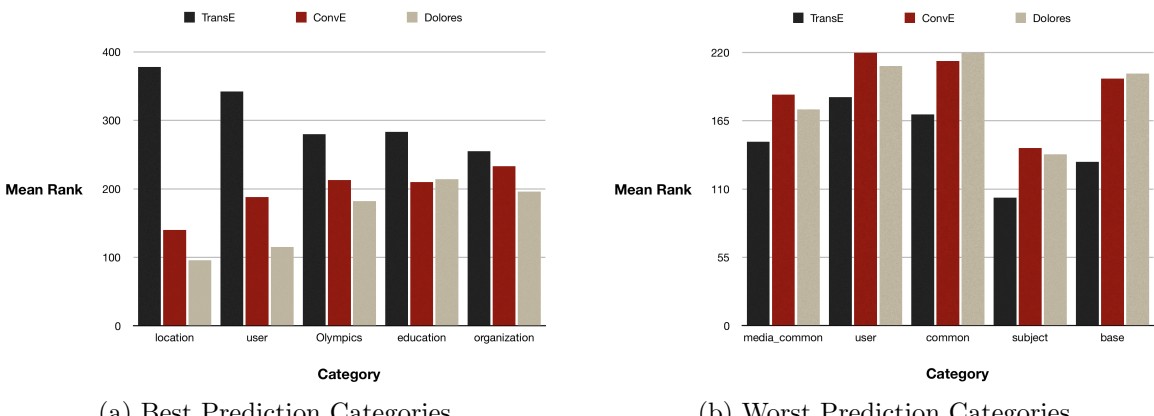

(a) Best Prediction Categories      (b) Worst Prediction Categories

Figure 3: Relation Categories with best and worst performance for Dolores in terms of mean rank (**lower is better**). Dolores performs exceedingly well on instances where the head entity is specific and tends to perform sub-optimally when the head entity is very generic and broad belonging to categories like "*base*" and "*common*". Please refer to Section 5 for detailed explanation and discussion.

Given a generic entity like "*Psychology*" ("*student-majoring*" being the relation), it is difficult to guess that a student named "*Yanni*" would be the expected tail entity. Generally, Dolores better models triples where the head entity is more specific compared to head entities which are broad and general.

## 6. Conclusion

In this paper, we introduce Dolores, a new paradigm for learning knowledge graph embeddings that learn not only contextual independent embeddings of entities and relations but also multiple context-dependent embeddings capturing different levels of abstraction, departing from the hitherto well-established paradigm of mapping entities and relations to vectors in $\mathcal{R}^d$ via a mapping function. By leveraging connections between 3 distinct fields: (a) large-scale network analysis, (b) natural language processing and (c) knowledge graphs, we learn rich knowledge graph embeddings that are deep and contextualized in contrast to prior models that are typically shallow. Besides, our learned embeddings can be easily incorporated into existing knowledge graph prediction models to significantly improve the performances on a variety of tasks.

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
