# OpenReview forum: "Dolores: Deep Contextualized Knowledge Graph Embeddings"
_AKBC.ws/2020/Conference — AKBC 2020_

### Official Review · AnonReviewer2 · 2020-03-26
**The paper shows good improvement over the existing work. The paper is interesting and the idea is novel.**

**Rating:** 8
**Confidence:** 4

**Review:**

This work proposes Dolores that captures contextual dependency between entity and relation pairs. The proposed method improves existing work. The idea of the paper is to incorporate Random walks and a language model. The idea is interesting and novel. However, the explanation of the training of the method is missing.

Comments
- The method has a shortcoming that it does not include the last entity. Let's assume that we have a sequence e_1,r_1,e_2,r_2,...,e_n,r_n,e_n+1. For the forward LSTM, e_n+1 is not included while e1 is not included for the backward LSTM.
- The loss function of the method is not defined.
- How to train the method is not clear. Is the method pretrained before each task?
- For the vector of Dolores, how are the multiple paths incorporated? In the experimental setting, the author generates 20 chains for each node. but how to incorporate the multiple chains is not clear
- For the link prediction task, it would be better to include ConvE + Dolores.

The paper shows good improvement over the existing work. The paper is interesting and the idea is novel.

---

### Official Review · AnonReviewer1 · 2020-03-29
**Bi-LSTM is proposed to learn knowledge graph embeddings. The key idea is not new and the evaluation has some flaws.**

**Rating:** 4
**Confidence:** 4

**Review:**

This paper presents a knowledge graph embedding approach that generates chains of entities and relations from a knowledge graph and learns the embeddings using Bi-LSTM. Results are shown to demonstrate that the proposed model can be incorporated into existing predictive models for different knowledge graph related tasks.

The key idea of using recurrent nets to learn embeddings from knowledge graph paths is not new. The authors try arguing the novelty, e.g., with respect to Das et al. [2017], in terms of 1) the different goal of learning generic embeddings rather than reasoning, and 2) the different way that paths are generated. However, the model by Das et al. also has a representation learning part; path generation of the proposed method cannot be seen as a contribution either as it is from the Node2Vec work.

In terms of the experimental evaluation, my main question is whether the comparison is fair. The authors compare original versions of existing methods with such methods incorporated with Dolores. This appears to be a comparison between a model without pretraining vs. the model with pretraining using Dolores. We all know that pretraining helps improve model performance and so, it is not surprising that a model incorporated with Dolores (e.g., ConvKB+Dolores) outperforms its original version and other comparison models without pretraining (e.g., Dolores, RNN-Path-entity). A more fair comparison would be comparing the effect of Dolores as a pretraining method with other pretraining methods.

Some technical details in the method and experiment sections need to be clarified:
- Section 3.4, "By accepting triples or paths from certain tasks as input (not the paths generated by path generator)" <= how exactly are paths obtained frm given tasks?
- How are the weights of embeddings at different layers learned in Eq. 4?
- In Section 4.1, it is mentioned that 20 chains are generated for each node. Is it always possible to extract 20 chains for any node? And, why is the parameter set to 20? How do different settings of this parameter affect the result?

---

### Official Review · AnonReviewer3 · 2020-03-31
**Representation learning yielding nice gains on a number of tasks. Some definitions could be clarified.**

**Rating:** 7
**Confidence:** 3

**Review:**

This paper presents a method of representing knowledge graph nodes and relations
by sampling paths and applying a sequence model. The approach is motivated by
recent advances in building contextualized word representations (in particular
ElMO) and the learned representations are applied to a number of downstream
tasks, with positive effects. This approach differs from other applications of
RNNs to path modeling in its focus on learning reusable representations by
modeling random walks, rather than attempting to learn to model paths of some
specific type.

The results are compelling. Dolores seems to yield representations that can be
applied effectively to a range of downstream tasks and models. I would like to
see more discussion of the models enhanced (ConvKB is introduced in the caption
of Table 3. only), and I would also like to see how much Dolores could improve
the non SOTA approaches. However, the current set of evaluations show that
Dolores provides significant gains over existing work in a number of settings.

Points for improvement:

I found the model description to be confusing. We are told repeatedly that the
approach is building representations of [entity, relation] pairs. It is not
clear from the description whether we are supposed to assume that the
representation of this pair is decomposed into separate, concatenated, entity
and relation components. From the description of the model, it seems that the
output layer applies a softmax over all possible (entity, relation)
pairs. Conversely, Figure 2 seems to illustrate a decomposition of the output
layer into concatenated entity and relation representations and Table 5
illustrates nearest neighbors of a single entity node (in context). Section 3
should be adapted to very explicitly state the nature of the predictive output
layer, and the loss that is used to train.


Since Dolores' training procedure is so different from that of the downstream
tasks, I would like to see some discussion of how the authors avoid overlap
between pre-training and test graphs for e.g. FB15k.

---

### Public Comment · ~Samuel_Broscheit1 · 2020-05-13
**Can you clarify the link prediction experiments?**

I just read your paper, and the results for link prediction on FB15K-237 are by far the best reported numbers on this dataset that I am aware of.

The paper does not describe how the link prediction works, so I would be very interested to understand it. My question is the following: As DOLORES are contextualized embeddings there must be a sentence/path during prediction, I wonder how you implemented that. My best guess ---based on what you wrote in the paper--- is the following example:

For a test triple

(Paris, cityIn, France) and the derived question (Paris, cityIn, ?)

you create a path

[CompanyInc, hasHeadquartIn, Paris, cityIn, France, countryIn, EU]

an then you feed the embeddings for [Paris, cityIn] and [France, countryIn] into ConvKB? And then you corrupt [France, countryIn] by replacing France with all other entities?

Or is it completely different? It would be great if you could give me some feedback on this!

---

> ### Author Response · Authors · 2020-05-18
> **Thanks for your comments!**
>
> Hi, thanks for your comments. There are indeed many ways of doing link prediction tasks and the way you described here is a feasible solution. We still need to make further refinements since there is some problem about the evaluation process of ConvKB-related methods that we overlooked. We will fix these issues soon in the camera-ready version.

---

### Decision · Program_Chairs · 2020-04-30

**Decision:**

Accept

**Comment:**

The paper introduces a simple and effective approach to obtaining entity embeddings (relying on RNN-encoded walks and ELMo style losses). The approach works well, is simple, and well-motivated.

While the underlying principles have been studied  (e.g., RNN embeddings of walks or learning representations relying on walks as in DeepWalk), there is enough novelty in the proposed method.  The other two reviewers are positive.

We would encourage the authors to address the reviewers' comments (e.g., regarding clarity in R3;  I had similar issues with understanding the model structure and the learning procedure  / objective).

It may be interesting to discuss the relation with graph neural networks (esp. with relational GCNs), which also learn a contextualized representation of entities, using similar types of losses. It may make sense to discuss why linearization can be beneficial (from representation learning or efficiency perspectives).